# Development and Method Validation of Butyric Acid and Milk Fat Analysis in Butter Blends and Blended Milk Products by GC-FID

**DOI:** 10.3390/foods11223606

**Published:** 2022-11-12

**Authors:** Arunee Danudol, Kunchit Judprasong

**Affiliations:** 1Department of Medical Sciences, Bureau of Quality and Safety of Food, Nonthaburi 11000, Thailand; 2Institute of Nutrition, Mahidol University, Nakhon Pathom 73170, Thailand

**Keywords:** butyric acid, butter fat, butter blends, blended milk products, GC-FID

## Abstract

Butyric acid is a short-chain saturated fatty acid with four carbon atoms in its molecule. It is unique to butter made from cow’s milk and is an indicator to evaluate the quality of butter and milk products as stated in their ingredient labels. This study determined the milk fat content of butter blends and blended milk products by analyzing the content of butyric acid prepared as derivatives of methyl esters prior to injection into a gas chromatography flame ionization detector (GC–FID). Results revealed that this method had specificity, a linear relationship for concentration in the range of 0.04–1.60 mg/mL, a coefficient of determination (R^2^) > 0.999, an instrumental limit of detection (LOD) and a limit of quantitative analysis (LOQ) at 0.01% and 0.10% of total fat, respectively, and an instrumental working range of 0.10–3.60% of total fat. The results of a precision study using relative standard deviation (RSD) was 1.3%, while an accuracy study using the spiking method showed % recovery in the range of 98.2–101.9%. The method linearity range for milk fat analysis had a good linear correlation in the range of 3–100% of total fat (R^2^ > 0.999). Results for method LOD and LOQ were 1% and 3% of total fat, respectively. This method also had good precision (1.3% RSD) and accuracy (99.6–100.1% recovery), which indicates reliability in terms of precision and accuracy. This method, therefore, can be used to check claims about the quality of blended butter and blended milk products to ensure consumer confidence in product quality.

## 1. Introduction

Butyric acid is a clear, colorless liquid with a pungent odor, a boiling point of 163.5 °C, and a chemical formula of CH_3_CH_2_CH_2_COOH or C_4_H_8_O_2_. Butyric acid is a type of saturated fatty acid containing only four carbon molecules (short-chain fatty acid, C4:0). These molecules occur naturally in the form of triglyceride compounds in the milk fat of cows and other ruminants but not in animal adipose or vegetable fats [1], which makes butyric acid an indicator or marker for milk fat blends [2].

Butter contains butyric acid at about 4% of total fatty acids [3]. It also contains up to 400 other different fatty acids [3,4]. Butyric acid is used in food and pharmaceutical industries as additives as well as in health foods. Studies report that butyric acid can inhibit the growth of cancer cells and fight against the formation of atherosclerosis [5,6,7,8]. Consequently, butter has a higher nutritional value than other fats and is classified within a group of important food products due to its nutritious nature, especially for infants, children, working-age people, pregnant women, and the elderly [9,10,11].

Global demand for butter in dairy products has continued to increase [9,12] since it is a component of many everyday foods, such as butter itself, blended butter, blended milk products, cheese, chocolate, flavored milk, cream, dietary supplements, baked goods, snacks, and other foods. Butter is also an important food trade item and, due to competition, has been imitated by using vegetable oils and other fats (e.g., margarine) that are classified as non-milk fats or non-dairy fat products [13]. These products have lower nutritional value, but their appearance is similar to butter. Moreover, their use in replacing real butter, in part or in whole, may not be clearly stated in a food product’s ingredient label, which can cause consumer confusion and misunderstanding. In addition, potential health risks arise because these substitutes are more difficult to digest in terms of their fat compared to butterfat products.

Producers and importers of blended butter and blended milk products [13] in countries such as the United States, New Zealand, Japan, the People’s Republic of China, and the European Union (EU) have recognized the problem of unfair trade competition in the price and quality of blended butter and blended milk products [14]. The International Food Standards Program, Codex Stan 256-2007, requires disclosure of ingredients, including those of butterfat products, on food product ingredient labels to make it clearer for consumers and reduce potential economic disadvantages. Consequently, each country has developed analytical methods for quality inspection.

A variety of methods exist to determine butyric acid and fat content that differ in terms of chemical use, materials, procedures, and experimental conditions of the methods, and the tools used for analysis. Analyzing butyric acid and fat content using a gas chromatography flame ionization detector (GC-FID) is the preferred method for laboratory testing [1,3,4,5,8,9,15,16,17,18], with most laboratories using it in a general laboratory setting [18]. Use of advanced instrumentation for butyric acid analysis, such as LC-APCI-MS/MS [17], high-resolution gas–liquid chromatograph (HR-GLC) [16], and Carbon-13 nuclear magnetic resonance (^13^C NMR spectroscopy), also have been reported [8]. Results using these instruments have relied on additional indicators other than butyric acid, such as triacylglycerols (TAGs, TGs). However, results have not been significantly different between GC-FID and advanced instrumentation.

As a producer and importer of blended butter and blended milk products, Thailand has taken steps to prevent the imitation of food products through the Thai Notification of the Ministry of Public Health (No. 348), B.E. 2555 Re: Margarine, blended butter, margarine products, and mixed milk products [19]. This Notification is based on the CODEX Alimentarius international food standards (CXS 256-2007 Standard for fat spreads and blended spreads [20]). It states that the quality of margarine and margarine products (fat spreads) must have milk fat not exceeding 3% of the total fat content and have a total fat content of 80–90% and 10–80% by weight, respectively. Blended butter and blended fat spreads must have a milk fat content of more than 3% of total fat and have a total fat content of 80–90% for blended butter and 10%, but less than 80% for blended fat spreads.

Milk fat content is an important indicator for butter blends and blended milk products with different fat contents depending on the type of product. Consequently, this present study centered on developing a method to determine fat content by analyzing butyric acid content from a butyric acid standard solution and a butter fat standard solution by adding valeric acid (internal standard) and preparing all solutions as a derivative of methyl ester. The reaction esterification allows easy vaporization of these fatty acids and is suitable for GC-FID analysis. The fat content is then analyzed by analogy with the content of butyric acid analyzed by standard curves. To our knowledge, the results of this new research concept have not been published previously. In addition, method validation of the new method was determined. This study’s output is to support the country’s laboratory mission to monitor the quality of fat in butter blends and blended milk products in order to facilitate consumer protection, prevent the adulteration of milk fat, and reduce potential damage to the country’s population and economy.

## 2. Materials and Methods

### 2.1. Standards and Chemicals

Butyric acid (C_4_H_8_O_2_, purity ≥ 99.6%) and valeric acid (C_5_H_10_O_2_, purity ≥ 99.9%) were purchased from Sigma-Aldrich^®^, Switzerland. Anhydrous Milk Fat (AMF, purity ≥ 99.95%) was purchased from Fonterra^®^, New Zealand. Petroleum ether (40–60 °C), methanol, n-heptane, sodium hydroxide, anhydrous sodium sulfate, and sodium chloride were analytical grade from RCI Labscan^®^, Thailand. Boron trifluoride-methanol (20% solution in methanol) was obtained from Merck^®^, Germany.

### 2.2. Instruments and Allied Equipment

Gas chromatography (GC, Perkin Elmer Clarus 600, Flame Ionization Detector [FID]) was used in this study. The analytical capillary column was SP^TM^ 2330 (30 m, 0.25 mm id, 0.2 μm film thickness). The GC condition for injection was: automatic liquid sampler injector, split mode 10:1, temperature set at 260 °C, and injection volume 1 µL. Oven temperature was programmed at initial 40 °C, held for 5 min, ramp rate 10 °C/min to 230 °C, and held for 3 min. Detector temperature was set at 260 °C with H_2_ flow rate of 45 mL/min and air zero flow rate of 450 mL/min, which was used to ignite the vapor to ionization. Carrier gas (He) was set at velocity of 20 cm/sec. Analysis time required for all separation stages was about 27 min.

Before analysis, a system suitability test was performed by injection, at 5 repetitions, using a standard solution (butyric acid, C4) and internal standard (valeric acid, C5) by preparing derivatives of methyl ester at a concentration of 0.40 mg/mL. Percentage relative standard deviation (%RSD) of retention time (RT) and peak area were calculated with acceptance criteria at RSD ≤ 1% and ≤ 2.5%, respectively.

### 2.3. Preparation of Standard Solutions

An internal standard solution (0.4 mg/mL) was prepared by weighing 100 mg of valeric acid, dissolving, and then adjusting by volume with methanol to 250 mL. A stock standard solution (2 mg/mL) was prepared by weighing 100 mg of butyric acid, dissolving, and then adjusting by volume with methanol to 50 mL. Working standard solutions at different concentrations (0.04, 0.08, 0.40, 0.80, 1.20, and 1.60 mg/mL) were prepared to create a matrix calibration curve by pipetting stock standard solution (2 mg/mL) at 0.1, 0.2, 1, 2, 3, and 4 mL, respectively, into flasks containing 0.2 g of palm oil (matrix blank). A standard solution of valeric acid (0.4 mg/mL) was pipetted (5 mL) into each flask to obtain a final concentration of 0.4 mg/mL.

Milk fat standard solutions were prepared at 8 concentrations (3, 5, 10, 20, 40, 60, 80, and 100% of total fat) to analyze butyric acid content and create a standard curve. They were prepared by weighing a standard substance (butter fat or anhydrous milk fat, AMF) mixed with palm oil to concentrations of butter fat at 3, 5, 10, 20, 40, 60, 80, and 100% of total fat. The prepared solution (0.2 g) and standard valeric acid solution (0.4 mg/mL, 5 mL) were added into each flask to a final concentration of 0.4 mg/mL.

### 2.4. Sample Preparation

Butter blends and blended milk products were randomly purchased from department stores (2–3 containers) to represent a total sample of approximately 200–300 g. They were mixed thoroughly, taking care in terms of liquid separation. The reserved samples were stored in a refrigerated compartment at 2–8 °C. To prepare each sample, fat was extracted from the sample by weighing 2 g of the prepared sample in a centrifuge tube (50 mL polypropylene) according to the standard method (ISO 17189/IDF 194, 2003: Butter, edible oil emulsions and spreadable fats—Determination of fat content [Reference method] [21]). Petroleum ether (20 mL) was added, mixed well with a vortex mixer, and shaken by hand for 1–2 min. Each sample was centrifuged at 3000 rpm for 5 min, after which the clear solution was poured into a beaker and evaporated in a water bath. The extraction processes were performed 2–3 times. The combined extracted fat was evaporated by blowing with N_2_ before drying in a hot air oven at 102 ± 2 °C for 2 h (to evaporate the residual moisture) and weighing to a constant weight (dried another 1 h in a hot air oven, weight difference less than 0.002 g). The lower weight value was used to calculate the amount of extracted fat.

Method blank was analyzed according to the above method without samples added. Matrix blank (palm oil) was also analyzed using all chemicals and processes. Due to the major constituents of both palm oil and butter being triglycerides of oil or fat, the analysis of butyric acid in the sample was analyzed only for oil/fat extraction. The prepared solutions were injected into the GC-FID machine to check for interferences. If the chromatogram had no interference peaks (only the peaks of butyric acid and valeric acid), the palm oil samples were used as matrix blanks for further testing of the method’s accuracy.

### 2.5. Analytical Method and Method Validation

To analyze butyric acid content, the extracted fat (0.2000–0.2005 g) was weighed into a 50 mL flat bottom flask. The standard solution of valeric acid (0.4 mg/mL) was added at 5 mL and then 4 mL of 0.5 N NaOH in methanol. The solution was connected to a condenser and refluxed for 8 min. Boron trifluoride-methanol (5 mL) was added via the condenser and boiled for 2 min. Organic solvent (n-heptane, 5 mL) was added via the condenser and boiled for 1 min. The flask was removed from the condenser, and 15 mL of saturated sodium chloride was added. The stopper was closed and shaken vigorously for approximately 15 sec. The saturated sodium chloride was continuously added until the n-heptane layer formed at the neck of the flask. The n-heptane layer, at approximately 2–3 mL, was transferred via a dropper into a test tube containing approximately 2 g of anhydrous sodium sulfate. This solution was mixed with a vortex mixer and left to stand for 5–10 min. The clear solution was then transferred into a vial and injected into the GC-FID for butyric acid determination.

Butyric acid analysis was performed in standard solution, butyric acid (working standard solution 6 concentrations), milk fat standard solution (8 concentrations), and sample solution according to Equation (1) below. Milk fat content was determined by creating a standard graph relationship between butyric acid content (percent of total fat, y-axis) and fat content (percent of total fat, x-axis) and then calculating the fat content from the linear equation according to Equation (2) below.
(1)$$\mathrm{Butyric}\;\mathrm{acid}\;\mathrm{content}\;\left(\%\;\mathrm{of}\;\mathrm{total}\;\mathrm{fat}\right)=\frac{C_{S}\times V_{S}}{W\times 10}$$
where C_S_ is the amount of butyric acid in the solution obtained from the standard curve (mg/mL); W is the weight of the extracted fat (g); V_S_ is the volume of the solution in the n-heptane layer (5 mL)
(2)$$\mathrm{Milk}\;\mathrm{fat}\;\mathrm{content}\;\left(\%\;\mathrm{of}\;\mathrm{total}\;\mathrm{fat}\right)=\frac{y-b}{m}$$

From a standard calibration graph, the relationship between butyric acid content (% of total fat) and fat content (% of total fat) is shown in linear equation as y = mx + b, where y is the amount of butyric acid (% of total fat), b is intercept, m is slope.

The linearity and working range of the method were assessed using standard butyric acid (working standard solution) at 6 concentration levels: 0.04, 0.08, 0.40, 0.80, 1.20, and 1.60 mg/mL. Each concentration was analyzed 3 times. Standard calibration curve was created between the peak area ratio of butyric acid and valeric acid and the butyric acid concentration. The determination of coefficient (R^2^) was calculated using the concentration at 8 levels (3, 5, 10, 20, 40, 60, 80, and 100% of total fat). Each concentration was analyzed 3 times. The determination of coefficient (R^2^) calculated the relationship between the amount of butyric acid (% of total fat) and milk fat content (% of total fat).

Limit of detection (LOD) and limit of quantitation (LOQ) of butyric acid content analysis were evaluated by adding the lowest concentration of standard butyric acid into a matrix blank and analyzed for 10 repetitions. LOD and LOQ were calculated from 3 times and 10 times of the standard deviation (SD), respectively. The accuracy and reliability of the LOQ value of both butyric acid and milk fat were verified by analyzing 10 times of LOQ concentration. The percentage of recovery (demonstrate accuracy) and the relative standard deviation (demonstrate precision) were assessed.

Accuracy and precision tests were performed with standard butyric acid at concentrations of 0.1, 0.8, 1.5, 2.2, 2.9, and 3.6% of total fat. Each concentration was analyzed for 10 repetitions and calculated in terms of the % recovery and the relative standard deviation of each concentration. The acceptance criteria for accuracy (% recovery) was 95–105%, and the acceptance criteria for precision using Horwitz’s equation. For milk fat at the concentrations of 3, 5, 10, 20, 40, 60, 80, and 100% of total fat, the acceptance criteria for accuracy (% recovery) was 98–102%, and the acceptance criteria for precision using Horwitz’s equation. For intermediate precision (analysis between days) study, blended milk products were analyzed twice a day for 10 days, and results were calculated using Analysis of variance (ANOVA) [22] as an output shown in Table 1. The intermediate precision (S_I_) is calculated from Equation (3).
(3)$$S_{I}=\sqrt{S_{r}^{2}+S_{\mathrm{between}}^{2}}$$
where $S_{r}=\sqrt{{\mathrm{MS}}_{w}}$, $S_{\mathrm{between}}$=$\sqrt{\frac{{\mathrm{MS}}_{b}-{\mathrm{MS}}_{w}}{n}}$, MS_b_ = mean square of between-group standard deviation, MS_w_ = mean square of within-group standard deviation.

Measurement uncertainty of the analytical method was estimated according to the Eurachem/CITAC guideline [23] by taking into account all uncertainty sources. The expanded uncertainty was reported at the 95% confidence level (coverage factor, k = 2).

## 3. Results

For the specificity of the method, the peak methyl ester of butyric acid (C4) and valeric acid (C5) in blended milk products were well-separated and showed symmetrical peaks, as in Figure 1. The retention times (Rt) of these two compounds were 5.07 and 7.26 min, respectively. No other fatty acid peaks were found to interfere with the interested peaks of samples.

For linearity and analytical range of the standard solution, all concentrations of butyric acid (0.04, 0.08, 0.40, 0.80, 1.20, and 1.6 0 mg/mL) provided a good relationship between the peak area ratio of butyric acid and valeric acid (C4/C5 ratio) and the concentration (Figure 2). A coefficient of determination (R^2^) was 0.9995, which was confirmed by the residual plot. The linearity for milk fat standard solutions at 3, 5, 10, 20, 40, 60, 80, and 100% of total fat was obtained (Figure 3). A standard calibration graph showed a good relationship between the concentration of butyric acid and the concentration of milk fat, with an R^2^ of 0.9993.

For the detection limit with a peak signal higher than that of signal to noise at three times, the limit of detection (LOD) for butyric acid was 0.01% of total fat and for milk fat was 1% of total fat. The limit of quantitative (LOQ) of butyric acid was 0.1% of total fat, and milk fat was 3% of total fat. The % recovery of butyric acid at concentrations of 0.1, 0.8, 1.5, 2.2, 2.9, and 3.6% of total fat was in the range of 98.2–101.9%, which met the AOAC standard guideline (97–103%) [24]. For the precision study assessing % RSDr, the range was 0.70–1.33%, as shown in Table 2. For milk fat fraction at concentrations 3, 5, 10, 20, 40, 60, 80, and 100% of total fat, the % recovery of all eight concentrations was in the range of 99.6–100.1%, which met the AOAC standard guideline (98–102%) [24] as shown in Table 2. The precision was in the range of 0.71–1.31% RSDr.

For intermediate precision (SI) from control sample analysis, two examples of mixed milk products with a concentration of butyric acid at 0.17 and 1.46% of total fat analyzed twice a day on 10 different days were analyzed by using one-way ANOVA statistic [22]. The results of the SI value presented as %RSD were 4.38 and 2.03%, respectively, which passed the acceptance criteria of the AOAC standard guideline [24] for reliability (Horwitz’s ratio; HORRAT < 2 and p-value > 0.05). This result indicated no significant difference between the two samples, as shown in Table 3a–c.

All sources of measurement uncertainty were included for estimating measurement uncertainty; for instance, repeatability of measurement, uncertainties due to calibration curve, analytical balances, volumetric flask, pipettes, and purity of standards. They were calculated into combined uncertainty (u_c_) and expanded uncertainty (U) at a 95% confidence level (k = 2). For example, the measurement uncertainty of butyric acid at 0.69% of total fat was 0.06% of total fat, and for milk fat, at 20% it was 2% of total fat (Table 4).

## 4. Discussion

The analytical method based on ISO 17678/IDF 202: 2010 (milk and milk products–determination of milk fat purity by gas chromatographic analysis of triglycerides, reference method) [25] analyzes milk fat purity through the analysis of triglycerides. However, milk fat contains up to 400 different types of fatty acids and has more than 50 atoms of carbon, which leads to a complicated method for determining milk fat. Analysts must be highly skilled in analyzing data and interpreting results correctly, making this method difficult in practice and analytically time-consuming. The International Food Standard, Codex Stan 256-2007 [20] (standard for fat spreads and blended spreads) recommends the AOAC official method (990.27: butyric acid in fats containing milk fat) [26] using gas chromatography flame ionization detector (GC-FID) for measurement. However, using a glass column for the analysis of butyric acid limits this method due to separation difficulties and a long conditioning time. To improve the separation efficiency of butyric acid and solvent peaks, adding % H_3_PO_4_ acid must be applied, which adversely affects peak symmetry and leads to instability. This method also has limitations in terms of milk fat analysis. Consequently, this present study aimed to address these issues through the analysis of both butyric acid and milk fat in butter blends and blended milk products. This new method could make laboratory analysis easier, faster, more efficient and accurate, and achieve more reliable results.

Extraction of butyric acid from total fats by method ISO 17189/IDF 194:2003 (Butter, edible oil emulsions, and spreadable fats—determination of fat content, Reference method) [21] provided only fats without nutrients and other substances. The fat extraction of this method was not fully validated. To improve accuracy and precision, this present study’s method used valeric acid as an internal standard for the analysis of butyric acid by GC-FID. Valeric acid had similar properties to the analyzed substance, provided a symmetry peak, clearly separated, and was stable, which can reduce variation and errors in the results. Some previous studies also chose valeric acid as the internal standard. Methyl esterification of butyric and valeric acids to methyl butyrate and methyl valerate provided efficiency of detection, no interference, and were well-separated with suitable retention times (5 and 7 min, respectively). In using palm oil as a matrix blank to prepare the working standard solution and milk fat standard solution, neither butyric acid nor valeric acid was found. All parameters of method validation of butyric acid content were performed according to the Eurachem Guideline [22] and Generation Accreditation Guidance [27]. Results showed that the developed method had a specificity and linearity test range from 0.04 to 1.60 mg/mL, which was sufficient for the analysis of samples. There was linearity in the standard calibration graph with the value of R^2^ 0.9995, a good limit of detection, and detection of quantitation. The results of this method also provided good precision and accuracy. Internal quality control using a control sample for analysis of each set of analyses provided good results. This method also participated in an external quality control (proficiency testing) program in Thailand, which achieved satisfactory results (|z| ≤ 2).

The analysis of milk fat content in blended milk fat using the results of butyric acid at each milk fat concentration led to a straight line in the calibration curve (calculate the value of the variable X or the fat content from the linear equation y = mx + b). This technique reduced the discrepancy of the result by more than one point on the average or representative value of the butyric acid content [14]. Results of the accuracy test for milk fat content revealed linearity (R^2^ 0.9993) with concentration ranging from 3% to 100% of total fat. The LOD and LOQ of milk fat were 1% and 3% of total fat, respectively, which is comparable to the CODEX STAN 256-2007 standard and conforms to the Thai Notification of the Ministry of Public Health (No. 348, B.E. 2555) [19] (Re: Margarine, blended butter, margarine products, and blended milk products) which states that milk fat must contain more than 3% of the total fat. For accuracy and precision tests, the developed method analyzed milk fat between 3 to 100% of total fat; acceptable recovery was achieved (99.6–100.1%) as well as acceptable precision (relative standard deviation of repeatability, RSD_r_ = 0.71–1.31%).

An assessment of the measurement uncertainty (MU) of the developed method was calculated from all sources of uncertainty, which can fulfill the requirements of international standard of testing laboratories ISO/IEC 17025: 2017 (Topic 5.4.6.2, the laboratory shall at least attempt to identify all the components of uncertainty and make a reasonable estimation). This method also provided measurement traceability to the SI unit and unbroken chain. Based on the butyric acid content at a concentration of 0.69% of total fat, the MU was reported at 0.06% of total fat (at a 95% confidence level) with 8.7% relative uncertainty, which met the criteria for consideration [28] which is less than two times of the predicted Horwitz’s relative standard deviation of reproducibility (RSD_R_). For the milk fat in blended milk at a concentration of 20.0% of total fat, the MU was 2% of total fat with 8.7% relative uncertainty. Considering all sources of measurement uncertainty, the source of uncertainty due to the calibration curve (C_0_) was a major part of the MU (up to 62%). The sources of uncertainty from repeatability (precision) and the volume were approximately the same percentage (about 13%). For the reduction of relative uncertainty of measurements, all three sources of uncertainty could be considered and reduced, especially uncertainty due to the calibration curve.

GC-FID is one of the instruments used in most analytical laboratories. The cost of a machine, running cost, and maintenance are approximately 8–10 times cheaper than high-technology instruments such as high-resolution gas–liquid chromatography (HR-GC) or gas chromatography–mass spectrometry. In this study, the analytical results of butyric acid showed no significant difference between GC-FID and HR-GC. A study by Joachim and Dietz [15] analyzed milk fat using the mean value of C4 content as a substitute for the milk fat calculation in the mixed fat sample that showed a relatively high error (± 10%). The result from actual C4 content showed a decrease in error (± 4%) at milk fat contents of 60% and 25%, and also did not study at a low amount of milk fat. In this study, the developed method determined the milk fat content from the actual milk fat contents as a calibration curve to control the tolerance to cover the milk fat content ranging from 3% to 100% of total fat. This method presented an error of less than 2% and can also analyze milk fat content as low as 3% that is consistent with or compatible with the CXS 256-2007 [20] benchmark with a milk fat content LOQ of 3%. Therefore, this study demonstrated the capability of GC-FID to analyze butyric acid and milk fat in butter blends and blended milk products with reliable results.

## 5. Conclusions

Development and method validation of butyric acid and milk fat analysis in butter blends and blended fat spreads by GC-FID is essential for laboratories that must conduct analyses for food production, quality control during production, and inspection tasks for the import and export of these food products. Due to incidents of food fraud and adulteration in both the quality of products and the prices of butter blends and blended milk products, the reliable measurement of butyric acid and milk fat is essential for all related stakeholders. This study demonstrated that the developed method provided reliable results according to international guidelines in terms of good specificity, linearity, LOD, LOQ, precision, accuracy, and measurement uncertainty. This method can also be used to analyze samples of other dairy products, such as butter, cheese, cream, and other fat products like margarine and margarine products. The outcome of this study could directly affect a country’s economy and mediate harmful effects on consumer health.

## Figures and Tables

**Figure 1 foods-11-03606-f001:**
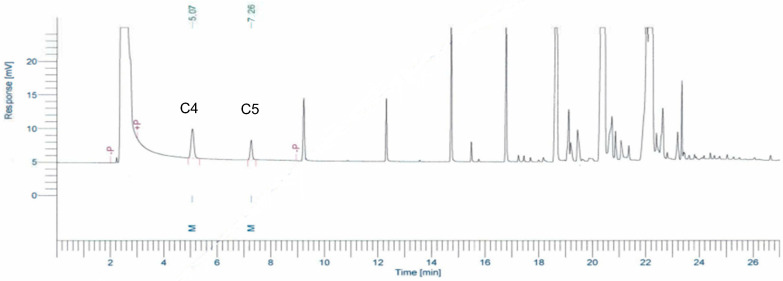
Chromatogram of butyric acid (C4) and valeric acid (C5) methyl esters in blended fat spreads analyzed by GC-FID.

**Figure 2 foods-11-03606-f002:**
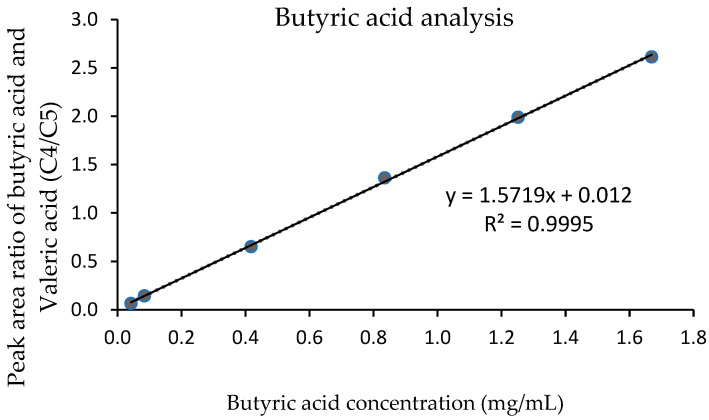
The standard calibration graph shows the relationship between the peak area ratio of butyric acid and valeric acid and the butyric acid concentration.

**Figure 3 foods-11-03606-f003:**
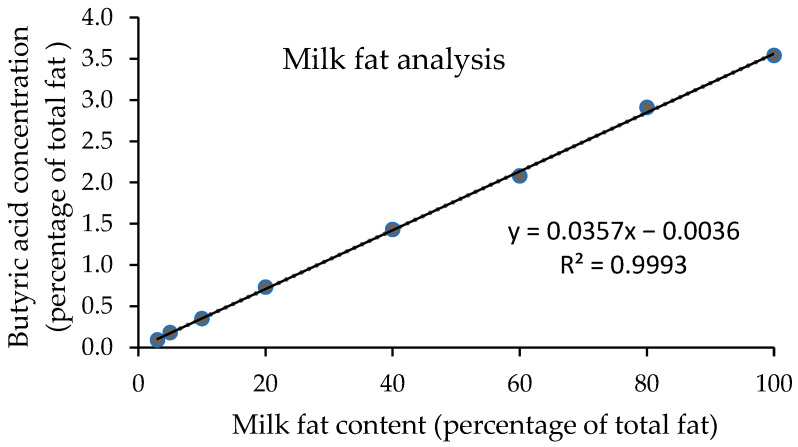
The standard calibration graph shows the relationship between butyric acid concentration and milk fat content.

**Table 1 foods-11-03606-t001:** One-way ANOVA output table, as example.

Source of Variations	Sum of Squares (SS)	ν	Mean Square (MS)	p	F_crit_
Between groups	SS_b_	p−1	MS_b_ = SS_b_/(p−1)	MS_b_/MS_w_	
Within groups (residuals)	SS_w_	N−p	MS_w_ = SS_w_/(N_p_)		
Total	SS_tot_ = SS_b_ + SS_w_	N−1			

**Table 2 foods-11-03606-t002:** Recovery study of butyric acid and milk fat content (n = 10).

Standard	Concentration (% of Total Fat)	% Recovery (Min–Max)	Mean ± SD
Butyric acid	0.1	98.22–101.89	99.91 ± 1.24
	0.8	98.49–101.51	99.81 ± 0.98
	1.5	98.25–101.40	99.80 ± 0.95
	2.2	98.56–101.68	100.13 ± 1.02
	2.9	98.97–101.03	100.12 ± 0.70
	3.6	98.89–101.72	100.02 ± 0.94
Milk fat	3	98.13–101.90	99.77 ± 1.31
	5	98.00–102.08	99.55 ± 1.23
	10	98.11–101.43	99.81 ± 1.14
	20	98.60–101.51	99.82 ± 0.97
	40	98.32–101.40	99.78 ± 1.02
	60	98.56–101.68	100.13 ± 1.02
	80	98.97–101.03	100.09 ± 0.71
	100	98.89–101.72	100.02 ± 0.94

**Table 3 foods-11-03606-t003:** (a). Results of butyric acid in milk products, analysis on different days, twice a day for 10 days. (b). Results of intermediate precision (SI) of butyric acid in sample 1 assessed by one-way ANOVA. (c). Results of intermediate precision (SI) of butyric acid in sample 2 assessed by one-way ANOVA.

**a**							
**Date**	**Example 1 (g/100 g Fat)**				**Example 2 (g/100 g Fat)**		
	**Replicate 1**		**Replicate 2**		**Replicate 1**	**Replicate 2**	
1	0.165		0.162		1.494	1.455	
2	0.184		0.173		1.435	1.438	
3	0.171		0.165		1.469	1.542	
4	0.169		0.185		1.406	1.399	
5	0.172		0.168		1.432	1.427	
6	0.185		0.178		1.420	1.455	
7	0.153		0.167		1.514	1.470	
8	0.175		0.182		1.458	1.503	
9	0.163		0.157		1.434	1.478	
10	0.186		0.170		1.511	1.465	
**b**							
**Source of Variation**	**SS**	**df**	**MS**	**F**	***p*-value**	**F Critical**	**S_I_**
Between Groups	0.0012	9	0.00013	2.663	0.0715	3.020	0.0074
Within Groups	0.0005	10	0.00005				
Total	0.0017	19					
**c**							
**Source of Variation**	**SS**	**df**	**MS**	**F**	***p*-value**	**F Critical**	**S_I_**
Between Groups	0.0196	9	0.00218	2.699	0.0689	3.020	0.0296
Within Groups	0.0081	10	0.00081				
Total	0.0277	19					

**Table 4 foods-11-03606-t004:** Estimation of measurement uncertainty of butyric acid and milk fat in butter blends.

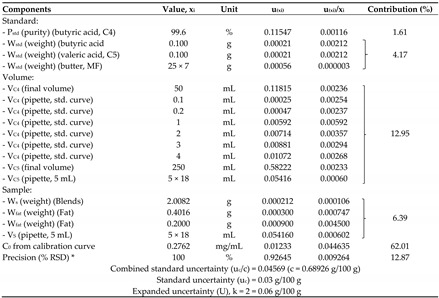

* % RSD was obtained from method validation (maximum RSD, 1.31%, standard uncertainty divided by 1.414).

## Data Availability

Data are contained within the article.

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
