# Peer review of "Development and Method Validation of Butyric Acid and Milk Fat Analysis in Butter Blends and Blended Milk Products by GC-FID"

_foods, 2022, doi:10.3390/foods11223606_

Round 1
Reviewer 1 Report
1. As the author mentioned, analyzing butyric acid and fat content using a gas chromatography flame ionization detector (GC-FID) is the preferred method for laboratory testing, so the advantages of this study should be compared with other reported studies.
2. Line 108, ALS?
3. Section 2.4, sample preparation, in this study, palm oil without butyric acid and valeric acid will be used as matrix blank, however, palm oil was totally different with butter or blended milk products, how to ensure the accuracy of the method.
4. Lines 172-173, “6 concentrations of milk fat standard solution, 8 concentrations and sample solution according to equation” should be reorganized.
5. In the equation (3), Sb and Sw should be specified.
6. In table 1, the last 3 line of the %Recovery of butyric acid and milk fat were the same data, is there a copy error?
Author Response
|
1. As the author mentioned, analyzing butyric acid and fat content using a gas chromatography flame ionization detector (GC-FID) is the preferred method for laboratory testing, so the advantages of this study should be compared with other reported studies. |
More information is added in the revised manuscript. - GC-FID is one of the instruments used in most analytical laboratories. The cost of a machine, running, and maintenance is approximately 8-10 times cheaper than high-technology instruments such as high-resolution-gas liquid chromatography (HR-GC) or gas chromatography-mass spectrometry. In this study, the analytical results of butyric acid showed no significant difference between GC-FID and HR-GC. - A study by Joachim & Dietz [15] analyzed milk fat using the mean value of C4 content as a substitute for the milk fat calculation in the mixed fat sample that showed a relatively high error (+ 10%). The result ​​from actual C4 content showed a decrease in error (+ 4%) at milk fat contents of 60% and 25%, and also did not study at a low amount of milk fat. In this study, the developed method determined the milk fat content from the actual milk fat contents as a calibration curve to control the tolerance to cover the milk fat content ranging from 3% to 100% of total fat. This method presented an error less than 2% and can also analyze milk fat content as low as 3% that is consistent with or compatible with the CXS 256-2007 benchmark with a milk fat content LOQ of 3%. |
|
2. Line 108, ALS? |
Full name of ALS is added in the revised manuscript as Automatic Liquid Sampler. |
|
3. Section 2.4, sample preparation, in this study, palm oil without butyric acid and valeric acid will be used as matrix blank, however, palm oil was totally different with butter or blended milk products, how to ensure the accuracy of the method. |
The major constituents of both palm oil and butter are triglycerides of oil or fat. The analysis of butyric acid in the sample was analyzed only for oil/fat extraction. This is why palm oil can use as matrix blank. This information was added into the revised manuscript. |
|
4. Lines 172-173, “6 concentrations of milk fat standard solution, 8 concentrations and sample solution according to equation” should be reorganized. |
This sentence is modified as “Butyric acid analysis was performed in standard solution, butyric acid (working standard solution 6 concentrations), milk fat standard solution (8 concentrations), and sample solution according to equation (1) below.” |
|
5. In the equation (3), Sb and Sw should be specified. |
The abbreviation of equation (3) are added as follow: MSb = mean square of between-group standard deviation MSw = mean square of within-group standard deviation |
|
6. In table 1, the last 3 line of the %Recovery of butyric acid and milk fat were the same data, is there a copy error? |
%Recovery of butyric acid and milk fat are obtained in the same value (not a copy error). |

Reviewer 2 Report
The manuscript describes the development of analytical procedure for butytic and valeric acids determination in butter. The study is well justified and well designed and needs clarification:
Why 102 Celcius degrees were applied for extract evaporation. BP of petroleum ether is below 60 degrees so such temperature is enough. 102 degrees can result in evaporation and losses of butyric acid which has BP of 163.
Author Response
|
The manuscript describes the development of analytical procedure for butyric and valeric acids determination in butter. The study is well justified and well designed and needs clarification:
Why 102 Celcius degrees were applied for extract evaporation. BP of petroleum ether is below 60 degrees so such temperature is enough. 102 Celcius degrees can result in evaporation and losses of butyric acid which has BP of 163.
|
After fat extraction and petroleum ether evaporated, butter or blended milk products still contained water or moisture of about 10-20%. It was necessary to evaporate the residual moisture at 102oC to allow only oil/fat to be further reacted by butyric acid. Butyric acid in the molecule of triglycerides will be stable (not be destroyed or lost). This information is added to the revised manuscript. |

Reviewer 3 Report
The method in the study is carefully described
However, it is not clear in what way it is new and unique. This must be clearly explained by the authors in the discussion and compared with the published works. Next, the authors must explain the fact that they use "standard milk fat" for evaluation. However, fat is the most variable nutrient of meca, and the proportion of fatty acids is variable depending on many factors. These facts need to be explained.
Otherwise, the claims and conclusions of the study are very bold.
Author Response
|
The method in the study is carefully described
However, it is not clear in what way it is new and unique. This must be clearly explained by the authors in the discussion and compared with the published works
Next, the authors must explain the fact that they use “standard milk fat” for evaluation, However, fat is the most variable nutrient of meca, and the proportion of fatty acids is variable depending on many factors. These facts need to be explained. Otherwise, the claims and conclusion of the study are very bold. |
- The development of analytical methods required a certified and reliable standard milk fat to test the validity of the method. The difference in the quality of milk fat should be checked for monitoring the quality and traced back to the different components and sources of production that may make the difference in results of milk fat. - A study by Joachim & Dietz [15] analyzed milk fat using the mean value of C4 content as a substitute for the milk fat calculation in the mixed fat sample that showed a relatively high error (+ 10%). The result ​​from actual C4 content showed a decrease in error (+ 4%) at milk fat contents of 60% and 25%, and also did not study at a low amount of milk fat. In this study, the developed method determined the milk fat content from the actual milk fat contents as a calibration curve to control the tolerance to cover the milk fat content ranging from 3% to 100% of total fat. This method presented an error less than 2% and can also analyze milk fat content as low as 3% that is consistent with or compatible with the CXS 256-2007 benchmark with a milk fat content LOQ of 3%. |

Round 2
Reviewer 1 Report
The paper can be accepted in present form.
Reviewer 3 Report
I have no further comment